# PUMPOUT: A META APPROACH FOR ROBUSTLY TRAINING DEEP NEURAL NETWORKS WITH NOISY LABELS

## ABSTRACT

It is challenging to train deep neural networks robustly on the industrial-level data, since labels of such data are heavily noisy, and their label generation processes are normally agnostic. To handle these issues, by using the memorization effects of deep neural networks, we may train deep neural networks on the whole dataset only the first few iterations. Then, we may employ early stopping or the small-loss trick to train them on selected instances. However, in such training procedures, deep neural networks inevitably memorize some noisy labels, which will degrade their generalization. In this paper, we propose a meta algorithm called *Pumpout* to overcome the problem of memorizing noisy labels. By using *scaled stochastic gradient ascent*, Pumpout actively squeezes out the negative effects of noisy labels from the training model, instead of passively forgetting these effects. We leverage Pumpout to upgrade two representative methods: MentorNet and Backward Correction. Empirical results on benchmark vision and text datasets demonstrate that Pumpout can significantly improve the robustness of representative methods.

## 1 INTRODUCTION

Learning from the industrial-level data is quite demanding, since labels of such data are heavily noisy (i.e., 50% of noisy labels), and their label generation processes are usually agnostic (Xiao et al., 2015; Jiang et al., 2018). Essentially, noisy labels of such data are corrupted from ground-truth labels without any prior assumptions (i.e., class-conditional noise (Natarajan et al., 2013)), which degrades the robustness of learning models. It is noted that industrial-level data is frequently emerging in our daily life, such as social-network data (Cha & Cho, 2012), E-commerce data (Xiao et al., 2015) and crowdsourcing data (Welinder et al., 2010; Han et al., 2018a).

Due to the large data volume, industrial-level data can be well handled by deep neural networks (Xiao et al., 2015). Thus, the key issue is how to train deep neural networks robustly on noisy labels of such data, since deep neural networks have the high capacity to fit noisy labels eventually (Zhang et al., 2017). To handle noisy labels, one common direction focuses on estimating the noise transition matrix (Goldberger & Ben-Reuven, 2017; Han et al., 2018b). For instance, Patrini et al. (2017) first leveraged a two-step solution to estimate the noise transition matrix. Based on this estimated matrix, they conducted backward loss correction, which is used for training deep neural networks robustly. However, the noise transition matrix is not easy to be estimated accurately, especially when the number of classes is large.

Motivated by the memorization effects of deep neural networks (Arpit et al., 2017), one emerging direction focuses on training only on selected instances (Jiang et al., 2018; Ren et al., 2018; Han et al., 2018c), which does not require any prior assumptions on noisy labels. Specifically, deep learning models are known to learn easy instances first, then gradually adapt to hard instances when training epochs become large (Arpit et al., 2017). Therefore, in the first few iterations, we may train deep neural networks on the whole dataset, and let them sufficiently learn clean instances in the noisy dataset. Then we may later conduct early stopping (Goodfellow et al., 2016), which tries to stop the training on noisy instances; or we may employ the small-loss trick (Jiang et al., 2018; Han et al., 2018c), which tries to perform the training selectively on clean (small-loss) instances.

However, when noisy labels indeed exist, no matter using early stopping or small-loss trick, deep networks inevitably memorize some noisy labels (Arpit et al., 2017), which will lead to the poor generalization performance (Zhang et al., 2017). In this paper, we design a meta algorithm called

*Pumpout*, which allows us to overcome the issue of memorizing noisy labels. The main idea of Pumpout is to actively squeeze out the negative effects of noisy labels from the training model, instead of passively forgetting these effects by further training. Specifically, on clean labels, Pumpout conducts stochastic gradient descent typically; while on noisy labels, Pumpout conducts *scaled stochastic gradient ascent*, instead of stopping gradient computation as usual. This aggressive policy can erase the negative effects of noisy labels actively and effectively.

We leverage Pumpout to upgrade two representative but orthogonal approaches in the area of "deep learning with noisy labels": MentorNet (Jiang et al., 2018) and Backward Correction (Patrini et al., 2017; van Rooyen & Williamson, 2018). We conducted experiments on benchmark vision and text datasets, namely simulated noisy *MNIST*, *CIFAR-10* and *NEWS* datasets. Empirical results demonstrated that, under both extremely noisy labels (i.e., 45% and 50% of noisy labels) and low-level noisy labels (i.e., 20% of noisy labels), the robustness of two upgraded approaches (by Pumpout) is obviously superior than that of original approaches.

## 2 PUMPOUT MEETS NOISY SUPERVISION

**Meta algorithm.** The original idea of Pumpout is to actively squeeze out the negative effects of noisy labels from the training model, instead of passively forgetting these effects. Intuitively, take "heart-broken" story as a supportive example. When you are disappointed in your former love, the best way to recover is to embrace the new love, which actively squeezes out the hurt from your mind, instead of repeatedly self-blaming. In the design of meta algorithm, we should consider how our meta algorithm can simultaneously benefit multiple orthogonal approaches in the area of deep learning with noisy labels, such as training on selected instances (Jiang et al., 2018), estimating the noise transition matrix (Patrini et al., 2017) and designing regularization (Miyato et al., 2016).

For this purpose, we generalize noisy labels into "not-fitting" labels, and generalize clean labels into "fitting" labels (details of the fitting condition will be discussed in Q1 below). In the high level, the meta algorithm Pumpout is to train deep neural networks by stochastic gradient descent on "fitting" labels, and train deep neural networks by *scaled stochastic gradient ascent* on "not-fitting" labels. In the low level, the proposed Algorithm 1 is named Pumpout. Specifically, we maintain deep neural network $f$ (with parameter $w_f$). When a single point $\{\mathbf{x}_i, y_i\}$ is sequentially selected from noisy set $\mathcal{D}$ (step 3), we first check whether $\{\mathbf{x}_i, y_i\}$ is fitting the discriminative condition or not. If yes, we conduct stochastic gradient descent typically (step 4); otherwise, we conduct scaled ($\gamma$) stochastic gradient ascent (step 5), which erases the negative effects of "not-fitting" labels. These "not-fitting" labels hinder us to train a robust model. The abstract algorithm arises three important questions.

---

**Algorithm 1** Meta Algorithm Pumpout.

---

1: **Input** network parameter $w_f$, learning rate $\eta$, maximum epoch $T_{\max}$, hyper parameter $0 \leq \gamma \leq 1$;
**for** $t = 1, 2, \ldots, T_{\max}$ **do**
  2: **Shuffle** training set $\mathcal{D}$;                                     //noisy dataset
  **for** $i = 1, \ldots, |\mathcal{D}|$ **do**
    3: **Select** $\{\mathbf{x}_i, y_i\}$ from $\mathcal{D}$ sequentially;
    **if** $\{\mathbf{x}_i, y_i\}$ *is fitting* **then**
      4: **Update** $w_f = w_f - \eta \nabla f(\mathbf{x}_i, y_i)$;            //stochastic gradient descent
    **end**
    **else**
      5: **Update** $w_f = w_f + \gamma \eta \nabla f(\mathbf{x}_i, y_i)$;      //scaled stochastic gradient ascent
    **end**
  **end**
**end**
6: **Output** $w_f$.

---

**Three important questions.**

Q1. What is the fitting condition?

Q2. Why do we need gradient ascent on non-fitting data, in addition to gradient descent on fitting data?

Q3. Why do we need to scale the stochastic gradient ascent on non-fitting data?

To answer the *first question*, we need to emphasize a view that orthogonal approaches require different fitting conditions. Intuitively, if a single point $\{\mathbf{x}_i, y_i\}$ satisfies a discriminative fitting condition, it means that our training model will regard this data point as a useful knowledge, and fitting on this point will benefit training the robust model. Conversely, if a single point $\{\mathbf{x}_i, y_i\}$ does not satisfy the discriminative fitting condition, it means that, our training model will regard this data point as useless knowledge, and want to erase the negative effects of this point actively. To instantiate the fitting condition, we provide two concrete cases in Algorithm 2 and Algorithm 3, respectively.

The above answer motivates our *second question*: why cannot we only conduct stochastic gradient descent on fitting data points (step 4). In other words, can we remove scaled stochastic gradient ascent (step 5) in Algorithm 1? In this case (removing step 5), our algorithm degenerates to training only on selected instances. However, once some of the selected instances are found to be false positives [1], our training model will fit on them, and thus the negative effects will inevitably occur (i.e., degrading the generalization (test accuracy)). Instead of passively forgetting these negative effects (i.e., further training over many epochs), we hope to actively squeeze out the negative effects from the training model by using scaled stochastic gradient ascent (step 5).

Lastly, the *third question* closely connects with the second one. Namely, why do we need *scaled* instead of *ordinary* stochastic gradient ascent? The answer can be intuitively explained. Assume that we view stochastic gradient ascent as correction to "not-fitting" labels, and view $0 \leq \gamma \leq 1$ as a scale parameter. When $\gamma = 1$, our Pumpout will squeeze out the negative effects with full fast rate; while when $\gamma = 0$, our Pumpout will not squeeze out any negative effects. Both cases are not optimal, since we empirically find that the best performance is usually chosen when $0 < \gamma < 1$ by using the validation set (Section 4). For the first case, the fast squeezing rate will negatively affect the convergence of our algorithm. For the second case, no squeezing rate will inevitably let deep neural networks memorize some "not-fitting" labels, which degrades their generalization (Zhang et al., 2017; Neyshabur et al., 2017).

## 3 PUMPOUT BENEFITS STATE-OF-THE-ART ALGORITHMS

In this section, we employ the idea of Pumpout to upgrade MentorNet and Backward Correction as follows. First, we briefly introduce the background of MentorNet and Backward Correction. Then, by using Pumpout, we propose upgraded MentorNet (Pumpout$_{SL}$) and upgraded Backward Correction (Pumpout$_{BC}$). Lastly, we explain the relations between MentorNet (Backward Correction) and Pumpout$_{SL}$ (Pumpout$_{BC}$).

### 3.1 UPGRADE MENTORNET

**MentorNet.** To handle noisy labels, an emerging direction focuses on training only on selected instances (Jiang et al., 2018; Ren et al., 2018; Han et al., 2018c), which is free of estimating the noise transition matrix, and also free of the class-conditional noise assumption (Natarajan et al., 2013). These works try to select clean instances out of the noisy ones, and then use them to update the network. Among those works, a representative method is MentorNet (Jiang et al., 2018), which employs the small-loss trick. Specifically, MentorNet pre-trains an extra network, and then uses the extra network for selecting small-loss instances as clean instances to guide the training. However, the idea of MentorNet is similar to the self-training approach (Chapelle et al., 2009), thus MentorNet inherits the same drawback of accumulated error caused by the sample-selection bias.

**Pumpout$_{SL}$.** Algorithm 2 represents the upgraded MentorNet using Pumpout approach (denoted as Pumpout$_{SL}$), where MentorNet uses the small-loss trick. Specifically, we maintain deep neural network $f$ (with parameter $w_f$). When a mini-batch $\bar{\mathcal{D}}$ is formed (step 3), we first let $f$ select a small proportion of instances in this mini-batch $\bar{\mathcal{D}}_s$ that have *small* training losses (step 4). The number of instances is controlled by $R(T)$, and $f$ only samples $R(T)$ percentage of instances out of the mini-batch. More importantly, we let $f$ select a proportion of instances in this mini-batch $\bar{\mathcal{D}}_b$ that have *big* training losses (step 5). The number of instances is controlled by $1 - R(T)$, and $f$ only samples

---

[1]`https://en.wikipedia.org/wiki/False_positives_and_false_negatives`

$1-R(T)$ percentage of instances out of the mini-batch. Then, we conduct stochastic gradient descent on small-loss instances $\bar{\mathcal{D}}_\text{s}$ (step 6); while we conduct scaled stochastic gradient ascent on big-loss instances $\bar{\mathcal{D}}_\text{b}$ (step 7), which actively erases the negative effects of big-loss instances. The update of $R(T)$ (step 8) follows Han et al. (2018c), in which extensive discussion has been conducted.

---

**Algorithm 2** Pumpout$_\text{SL}$. The fitting condition is whether a point belongs to small-loss instances.

---

1: **Input** network parameter $w_f$, learning rate $\eta > 0$, estimated noise rate $\tau$, maximum epoch $T_\text{max}$, maximum iteration $N_\text{max}$, hyper parameter $0 \le \gamma \le 1$;

**for** $T = 1, 2, \ldots, T_\text{max}$ **do**

    2: **Shuffle** training set $\mathcal{D}$;                                          //noisy dataset

    **for** $N = 1, \ldots, N_\text{max}$ **do**

        3: **Draw** mini-batch $\bar{\mathcal{D}}$ from $\mathcal{D}$;

        4: **Sample** $\bar{\mathcal{D}}_\text{s} = \arg\min_{\bar{\mathcal{D}}} \ell(f, \bar{\mathcal{D}}, R(T))$;        //sample $R(T)\%$ small-loss instances

        5: **Sample** $\bar{\mathcal{D}}_\text{b} = \arg\max_{\bar{\mathcal{D}}} \ell(f, \bar{\mathcal{D}}, 1 - R(T))$;    //sample $1 - R(T)\%$ big-loss instances

        6: **Update** $w_f = w_f - \eta\nabla f(\bar{\mathcal{D}}_\text{s})$;        //update $w_f$ by stochastic gradient descent on $\bar{\mathcal{D}}_\text{s}$;

        7: **Update** $w_f = w_f + \gamma\eta\nabla f(\bar{\mathcal{D}}_\text{b})$; //update $w_f$ by scaled stochastic gradient ascent on $\bar{\mathcal{D}}_\text{b}$;

    **end**

    8: **Update** $R(T) = 1 - \min\left\{\frac{T}{T_k}\tau, \tau\right\}$;

**end**

9: **Output** $w_f$.

---

**Relations between MentorNet and Pumpout$_\text{SL}$.** If we remove step 5 and step 7 in Algorithm 2, Pumpout$_\text{SL}$ algorithm will be reduced to the core version of MentorNet, namely self-paced MentorNet. It means that Pumpout$_\text{SL}$ algorithm is more aggressive than MentorNet in essence. Namely, Pumpout$_\text{SL}$ conducts not only stochastic gradient descent on small-loss instances (like MentorNet), but also scaled stochastic gradient ascent on big-loss instances.

## 3.2   Upgrade Backward Correction

**Backward Correction and its non-negative version.** To handle noisy labels, the other popular direction focuses on estimating the noise transition matrix (Goldberger & Ben-Reuven, 2017; Patrini et al., 2017; Han et al., 2018b). Among those works, a representative method is Backward Correction. Specifically, Patrini et al. (2017) leveraged a two-step solution to estimate the noise transition matrix heuristically. Then they employed the estimated matrix to correct the original loss, and robustly train a deep neural network based on the new loss function.

**Theorem 1** *(Backward Correction, Theorem 1 in (Patrini et al., 2017)) Suppose that the noise transition matrix* $\mathbf{T}$ *is non-singular, where* $\mathbf{T}_{ij} = \Pr(\tilde{y} = j|y = i)$ *given that noisy label* $\tilde{y} = j$ *is flipped from clean label* $y = i$. *Given loss* $\ell$ *and network parameter* $w_f$, *Backward Correction is defined as*

$$\ell^{\leftarrow}(\mathbf{x}, y; w_f) = \mathbf{T}^{-1}\ell(\mathbf{x}, y; w_f). \tag{1}$$

*Then, corrected loss is unbiased, namely,*

$$\mathbb{E}_{\tilde{y}|\mathbf{x}}\ell^{\leftarrow}(\mathbf{x}, y; w_f) = \mathbb{E}_{y|\mathbf{x}}\ell(\mathbf{x}, y; w_f), \forall\mathbf{x}. \tag{2}$$

**Remark 1** *Backward Correction operates on the loss vector directly. It is unbiased. LHS of Eq. (2) draws from noisy labels, and RHS of Eq. (2) draws from clean labels. Note that the corrected loss is differentiable, but not always non-negative (van Rooyen & Williamson, 2018).*

If the model being trained is flexible, such as a deep neural network, the backward loss correction will lead to negative risks, and the hazardous aspect is to yield an over-fit issue. Motivated by Kiryo et al. (2017), we should conduct a non-negative correction again based on the backward-corrected loss. The reason is that the risk should always be greater than 0 or equal to (Kiryo et al., 2017).

**Theorem 2** *(Non-negative Backward Correction) Suppose that the noise transition matrix* $\mathbf{T}$ *is non-singular, where* $\mathbf{T}_{ij} = \Pr(\tilde{y} = j | y = i)$ *given that noisy label* $\tilde{y} = j$ *is flipped from clean label* $y = i$. *Given loss* $\ell$ *and network parameter* $w_f$, *Non-negative Backward Correction is defined as*

$$\ell_m^{\leftarrow}(\mathbf{x}, y; w_f) = \max\{0, \mathbf{1}^{\top}\mathbf{T}^{-1}\ell(\mathbf{x}, y; w_f)\}, \tag{3}$$

*where* $\mathbf{1}_{k \times 1}$. *Then, the corrected loss* $\ell_m^{\leftarrow}(\mathbf{x}, y; w_f)$ *is non-negative.*

**Remark 2** $\ell_m^{\leftarrow}(\mathbf{x}, y; w_f)$ *is a non-negative scalar. Our key claim is to overcome the over-fit issue by non-negative correction.*

However, the above non-negative correction is passive, since $\max$ operator means stopping gradient computation on negative-risk instances. This correction may not achieve the optimal performance. Namely, when $\mathbf{1}^{\top}\mathbf{T}^{-1}\ell(\mathbf{x}, y; w_f) \geq 0$, we conduct stochastic gradient descent; otherwise, we do not perform the operation of stochastic gradient. To propose an aggressive non-negative correction, we reverse the gradient computation at negative-risk instances. Specifically, we use the Pumpout approach to improve Non-negative Backward Correction. Namely, when $\mathbf{1}^{\top}\mathbf{T}^{-1}\ell(\mathbf{x}, y; w_f) \geq 0$, we conduct stochastic gradient descent typically; when $\mathbf{1}^{\top}\mathbf{T}^{-1}\ell(\mathbf{x}, y; w_f) \leq 0$, we conduct scaled stochastic gradient ascent. This brings our Algorithm 3.

**Pumpout$_{\text{BC}}$.** Algorithm 3 represents the upgraded Backward Correction using the Pumpout approach (denoted as Pumpout$_{\text{BC}}$), where Backward Correction is defined in Theorem 1. If the model being trained is flexible (i.e., a deep neural network), Backward Correction will lead to negative risks (Patrini et al., 2017), which subsequently yields an over-fit issue. To mitigate this issue, we maintain deep neural network $f$ (with parameter $w_f$). When a single point $\{\mathbf{x}_i, y_i\}$ is sequentially selected from the $j$-th mini-batch $\bar{\mathcal{D}}$ (step 5), we first compute the temporary gradient $g_t$ at this point (step 6). If Backward Correction produces a positive risk at this point, namely $\mathbf{1}^{\top}\mathbf{T}^{-1}\ell(\mathbf{x}_i, y_i; w_f) \geq \beta \geq 0$ (definitions of $\mathbf{T}$ and $\ell$ are in Theorem 1), we accumulate gradient $G_a$ by the gradient descent (step 7); otherwise, we accumulate gradient $G_a$ by the scaled gradient ascent (step 8), and this step erases the negative effects of negative-risk instances. Lastly, we average the accumulated gradient (step 9) and update parameter $w_f$ by stochastic optimization (step 10).

---

**Algorithm 3** Pumpout$_{\text{BC}}$. The fitting condition is whether a point satisfies $\mathbf{1}^{\top}\mathbf{T}^{-1}\ell(\mathbf{x}_i, y_i; w_f) \geq \beta$.

---
1. **Input** network parameter $w_f$, learning rate $\eta > 0$, maximum epoch $T_{\max}$, hyper parameter $\beta \geq 0$ and $0 \leq \gamma \leq 1$;

**for** $T = 1, 2, \ldots, T_{\max}$ **do**

   2. **Shuffle** training set $\mathcal{D}$ into $n$-mini batches with batch size $k$;         //noisy dataset

   **for** $j = 1, \ldots, n$ **do**

      3. **Reset** $G_a = 0$;         //gradient accumulator

      4: **Draw** $j$-th mini-batch $\bar{\mathcal{D}}$ from $\mathcal{D}$;

      **for** $i = 1, \ldots, k$ **do**

         5. **Select** $\{\mathbf{x}_i, y_i\}$ from $\bar{\mathcal{D}}$ as $i$-th data point;

         6. **Set** $g_t = \nabla_{w_f}\{\mathbf{1}^{\top}\mathbf{T}^{-1}\ell(\mathbf{x}_i, y_i; w_f)\}$;         //temp gradient

         **if** $\mathbf{1}^{\top}\mathbf{T}^{-1}\ell(\mathbf{x}_i, y_i; w_f) \geq \beta$ **then**

            7. **Update** $G_a = G_a + g_t$;         //gradient descent

         **end**

         **else**

            8. **Update** $G_a = G_a - \gamma g_t$;         //scaled gradient ascent

         **end**

      **end**

      9. **Average** $g_a = G_a / k$;

      10. **Update** $w_f = w_f - \eta g_a$;         //stochastic optimization

   **end**

**end**

11. **Output** $w_f$.

---

**Relations between Non-negative Backward Correction and Pumpout$_{\text{BC}}$.** If we remove line 8 in Algorithm 3, Pumpout$_{\text{BC}}$ algorithm will be reduced to Non-negative Backward Correction. It means Pumpout$_{\text{BC}}$ algorithm is an aggressive version of Non-negative Backward Correction. Namely, Pumpout$_{\text{BC}}$ conducts not only stochastic gradient descent on nonnegative-risk instances, but also scaled stochastic gradient ascent on negative-risk instances to erase their negative effects.

# 4 EXPERIMENTS

**Datasets.** We verify the effectiveness of our Pumpout approach on three benchmark datasets, including two vision datasets and one text dataset. *MNIST*, *CIFAR-10* and *NEWS* are used here (Table 1), as these data sets are popularly used for evaluation of noisy labels in the literature (Reed et al., 2015; Goldberger & Ben-Reuven, 2017; Patrini et al., 2017; Kiryo et al., 2017).

Table 1: Summary of data sets used in the experiments.

|  | # of training | # of testing | # of class | size of image/text |
|---|---|---|---|---|
| *MNIST* | 60,000 | 10,000 | 10 | 28×28 |
| *CIFAR-10* | 50,000 | 10,000 | 10 | 32×32 |
| *NEWS* | 11,314 | 7,532 | 2 | 300-D |

Since all datasets are clean, following (Reed et al., 2015; Patrini et al., 2017), we need to corrupt these datasets manually by the noise transition matrix $\mathbf{T}$, where $\mathbf{T}_{ij} = \Pr(\tilde{y} = j | y = i)$ given that noisy $\tilde{y}$ is flipped from clean $y$. Assume that the matrix $\mathbf{T}$ has two representative structures (Figure 1): (1) Pair flipping (Han et al., 2018b): a real-world application is the fine-grained classification, where you may make mistake only within very similar classes in the adjunct positions; (2) Symmetry flipping (Van Rooyen et al., 2015). Their precise definition is in Appendix A.

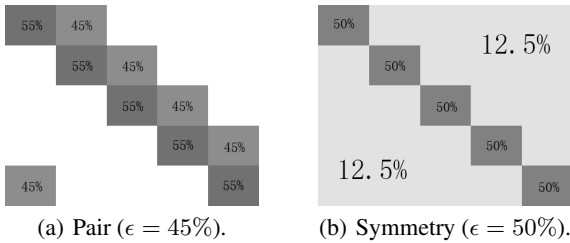

(a) Pair ($\epsilon = 45\%$).  (b) Symmetry ($\epsilon = 50\%$).

Figure 1: Transition matrices of different noise types (using 5 classes as an example).

This paper first verifies whether Pumpout can significantly improve the robustness of representative methods on *extremely* noisy supervision, the noise rate $\epsilon$ is chosen from $\{0.45, 0.5\}$. Intuitively, this means almost half of the instances have noisy labels. Note that, the noise rate $> 50\%$ for pair flipping means over half of the training data have wrong labels that cannot be learned without additional assumptions. In addition to extremely noisy settings, we also verify whether Pumpout can significantly improve the robustness of representative methods on *low-level* noisy supervision, where $\epsilon$ is set to 0.2. Note that pair case is much harder than symmetry case. In Figure 1(a), the true class only has 10% more correct instances over wrong ones. However, the true has 37.5% more correct instances in Figure 1(b). Meanwhile, similarly to (Reed et al., 2015; Goldberger & Ben-Reuven, 2017; Jiang et al., 2018), we did not make any implicit assumption behind Pumpout.

**Baselines.** To verify the efficacy of Pumpout, we compare two orthogonal approaches in deep learning with noisy labels. The first set (SET1) comparison is to check whether Pumpout can improve the robustness of MentorNet. (i) MentorNet (Jiang et al., 2018). (ii) Pumpout$_{\text{SL}}$ (Algorithm 2). The second set (SET2) comparison is to check whether Pumpout can improve the robustness of Backward Correction. (i) Backward Correction (Patrini et al., 2017) (denoted as "BC", Theorem 1). (ii). Non-negative backward correction (denoted as "nnBC", Theorem 2). (iii) Pumpout$_{\text{BC}}$ (Algorithm 3). As a simple baseline, we also compare with the standard deep neural network that directly learns on the noisy training set (denoted as "Standard"). Note that, the choice of two baselines is to justify whether Pumpout can benefit representative state-of-the-art algorithms. The readers are encouraged to upgrade other methods, such as Reed et al. (2015), Goldberger & Ben-Reuven (2017), and Kiryo et al. (2017) by using Pumpout.

For the fair comparison, we implement all methods with default parameters by PyTorch, and conduct all the experiments on a NVIDIA K80 GPU. Standard CNN is used with Leaky ReLU (LReLU) activation function (Maas et al., 2013); ResNet is used with ReLU activation function; and MLP is used with Softsign activation function (Glorot & Bengio, 2010). The detailed architectures are in Appendix B. Namely, we used the 9-layer CNN (Miyato et al., 2016; Laine & Aila, 2017) with dropout and batch-normalization for *MNIST*, ResNet-32 (He et al., 2016) with batch-normalization for *CIFAR-10*, and 3-layer MLP (Kiryo et al., 2017) with batch-normalization for *NEWS*, since the network structures we used here are standard test bed for weakly-supervised learning. For all datasets, Adam optimizer (momentum=0.9) with an initial learning rate of 0.001, the batch size is set to 128 and runs for 200 epoch. Note that, the focus of our paper is to explore the efficacy of Pumpout. Therefore, we use Adam optimizer in all experiments for fair comparison without using data augmentation trick (Zhang & Sabuncu, 2018; Ma et al., 2018).

**Experimental setup.** For *SET1*, the most important parameter of our Pumpout$_{SL}$ and MentorNet is $R(T)$. Here, we assume the noise level $\epsilon$ is known and set $R(T) = 1 - \tau \cdot \min(T/T_k, 1)$ with $T_k = 10$ and $\tau = \epsilon$. If $\epsilon$ is not known in advanced, $\epsilon$ can be inferred using validation sets (Liu & Tao, 2016). The choices of $R(T)$ and $\tau$ follows Han et al. (2018c). Note that $R(T)$ only depends on the memorization effect of deep networks but not any specific datasets. For *SET2*, the most important parameters of our Pumpout$_{BC}$ and nnBC are $\beta$ and $\gamma$ respectively. Specifically, the degree of tolerance is controlled by $\beta$ ($\beta \geq 0$), and the scale of gradient ascent is controlled by $\gamma$ ($0 \leq \gamma \leq 1$). The choices of $\beta$ and $\gamma$ follows Kiryo et al. (2017). Namely, for $\beta$, we directly set it to zero and $\gamma$ is chosen among $\{0, 0.001, 0.005, 0.01, 0.05, 0.1, 0.5, 1\}$ via a validation set.

This paper provides two upgraded approaches to train deep neural networks robustly under noisy labels. Thus, our goal is to classify the clean instances as accurately as possible, and the measurement for both *SET1* and *SET2* is the test accuracy, i.e., *test accuracy = (# of correct predictions) / (# of test dataset)*. Besides, for *SET1*, we also use the label precision in each mini-batch, i.e., *label precision = (# of clean labels) / (# of all selected labels)*. Specifically, we sample $R(T)$ of small-loss instances in each mini-batch, and then calculate the ratio of clean labels in the small-loss instances. Intuitively, higher label precision means less noisy instances in the mini-batch after sample selection; and the algorithm with higher label precision is also more robust to the label noise. All experiments are repeated five times. In each figure, the error bar for standard deviation is highlighted as a shade.

Before delving into Section 4.1 and 4.2, there are two important points to be emphasized. First, the memorization effects of deep networks (Arpit et al., 2017) means that standard deep networks will fit clean instances first, then overfit noisy instances gradually. These effects will inevitably degrade the generalization performance (i.e., test accuracy). Second, Pumpout$_{SL}$ may not suffer from or greatly alleviate the accumulated error in MentorNet, since Pumpout$_{SL}$ actively squeezes out the negative effects of noisy labels from the training model, instead of passively forgetting these effects.

## 4.1 RESULTS OF PUMPOUT$_{SL}$ AND MENTORNET

***MNIST.*** In Figure 2, we show test accuracy (top) and label precision (bottom) vs number of epochs on *MINIST* dataset. In all three plots, we can clearly see the memorization effects of deep networks (Arpit et al., 2017), i.e., test accuracy of Standard first reaches a very high level and then gradually decreases. Thus, a good robust training method should stop or alleviate the decreasing process. On this point, our Pumpout$_{SL}$ almost stops the decreasing process in the easier Symmetric-50% and Symmetric-20% cases. Meanwhile, compared to MentorNet, our Pumpout$_{SL}$ alleviates the decreasing process in the hardest Pair-45% case. Thus, Pumpout$_{SL}$ consistently achieves the higher accuracy over MentorNet.

To explain such good performance, we plot label precision (bottom). Compared to Standard, we can clearly see that both Pumpout$_{SL}$ and MentorNet can successfully pick clean instances out. However, our Pumpout$_{SL}$ achieves the higher label precision on not only the easier Symmetric-50% and Symmetric-20% cases, but also the hardest Pair-45% case. This shows our approach is better at finding clean instances due to the usage of scaled stochastic gradient ascent.

***CIFAR-10.*** Figure 3 shows test accuracy and label precision vs number of epochs on *CIFAR-10* dataset. Again, on test accuracy, we can see Pumpout$_{SL}$ strongly stops the memorization effects of deep networks. More importantly, on the easier Symmetric-50% and Symmetric-20% cases, it works better and better along with the training epochs. On label precision, while Standard fails to

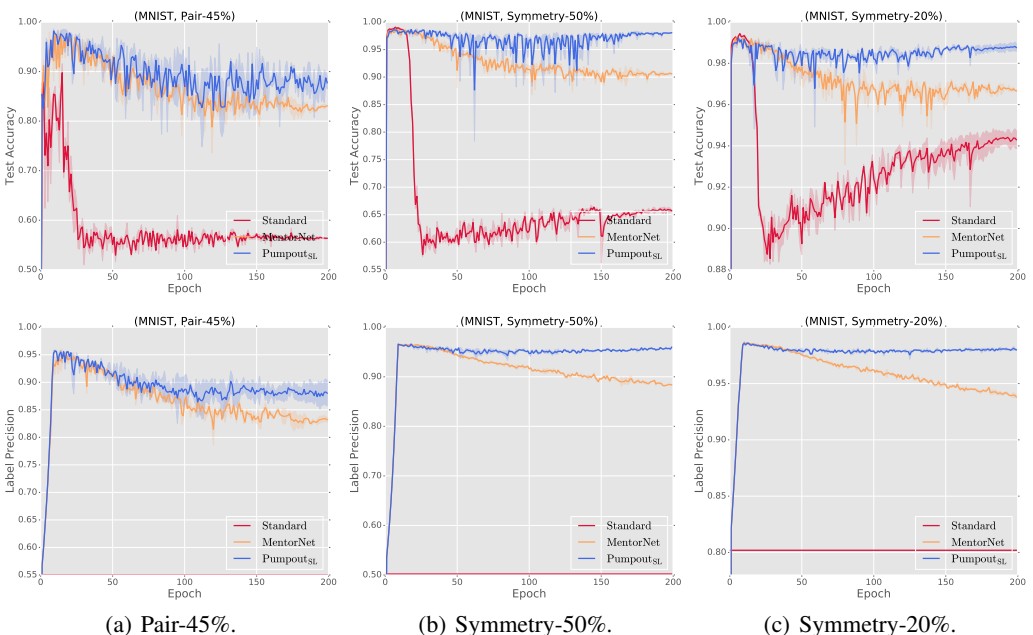

Figure 2: Results of Pumpout$_{SL}$ and MentorNet on *MNIST* dataset. Top: test accuracy vs number of epochs; bottom: label precision vs number of epochs.

find clean instances, both Pumpout$_{SL}$ and MentorNet can do this. However, due to the usage of scaled stochastic gradient ascent, Pumpout$_{SL}$ is stronger and find more clean instances.

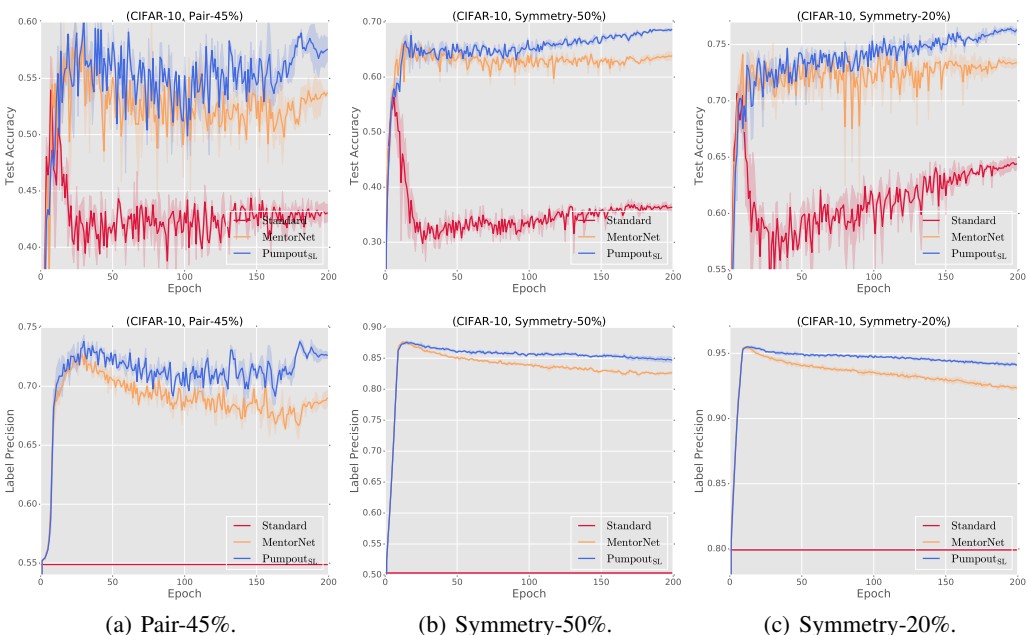

Figure 3: Results of Pumpout$_{SL}$ and MentorNet on *CIFAR-10* dataset. Top: test accuracy vs number of epochs; bottom: label precision vs number of epochs.

***NEWS.*** Figure 4 shows test accuracy and label precision vs number of epochs on *NEWS* dataset. On test accuracy, we can see Pumpout$_{SL}$ stops the memorization effects of deep networks to some degree. Especially on the harder Pair-45% and Symmetric-50% cases, Pumpout$_{SL}$ obviously achieves the higher accuracy over MentorNet along with the training epochs. On label precision, while Stan-

dard fails to find clean instances again, Pumpout$_{SL}$ can achieve this especially on the hardest case due to the usage of scaled stochastic gradient ascent.

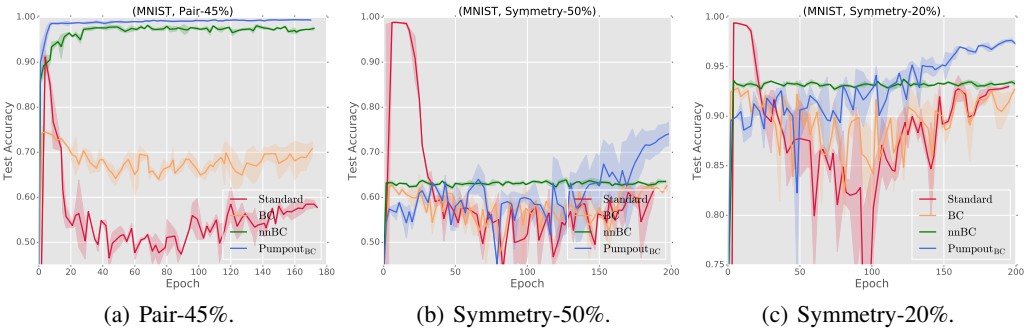

Figure 4: Results of Pumpout$_{SL}$ and MentorNet on *NEWS* dataset. Top: test accuracy vs number of epochs; bottom: label precision vs number of epochs.

## 4.2 RESULTS OF PUMPOUT$_{BC}$ AND NNBC

***MNIST.*** Figure 5 shows test accuracy vs number of epochs on *MNIST* dataset. In all three plots, we can see the memorization effects of deep networks, i.e., test accuracy of Standard first reaches a very high level and then gradually decreases. However, our Pumpout$_{BC}$ fully stops the decreasing process in the hardest Pair-45% case. Meanwhile, in the easier Symmetric-50% and Symmetric-20% cases, our Pumpout$_{BC}$ works better and better along with the training epochs though it fluctuates. Moreover, our Pumpout$_{BC}$ finally achieves the higher accuracy over both BC and nnBC.

Figure 5: Results of Pumpout$_{BC}$ and nnBC on *MNIST* dataset. Test accuracy vs number of epochs.

***CIFAR-10.*** Figure 6 shows test accuracy vs number of epochs on *CIFAR-10* dataset. Again, in all three plots, we can see the memorization effects of deep networks. However, in the hardest Pair-45% case and the easiest Symmetry-20% case, our Pumpout$_{BC}$ overcomes this issue and works better and better along with the training epochs though it fluctuates slightly. Specifically, in the hardest case, our Pumpout$_{BC}$ obviously achieves the higher accuracy over both BC and nnBC. Meanwhile, in the Symmetric-50% case, our Pumpout$_{BC}$ becomes comparable with other methods.

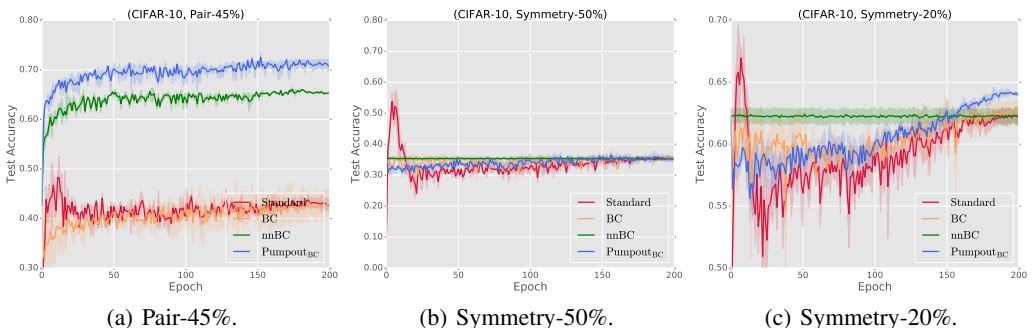

(a) Pair-45%.          (b) Symmetry-50%.          (c) Symmetry-20%.

Figure 6: Results of Pumpout_BC and nnBC on *CIFAR-10* dataset. Test accuracy vs number of epochs.

***NEWS.*** Figure 7 shows test accuracy vs number of epochs on *NEWS* dataset. In all three plots, we can see the memorization effects of deep networks again. However, our Pumpout_BC fully stops the decreasing process in two harder Pair-45% and Symmetry-50% cases, and alleviates the decreasing process in one easier Symmetry-20% case. Meanwhile, in the hardest Pair-45% case, our Pumpout_BC works better and better along with the training epochs. In this hardest case, our Pumpout_BC finally achieves the higher accuracy over both BC and nnBC, although its accuracy falls behind BC in the first 90 epochs and nnBC in the first 50 epochs. Besides, in two symmetry cases, our Pumpout_BC obviously achieves the higher accuracy over both BC and nnBC.

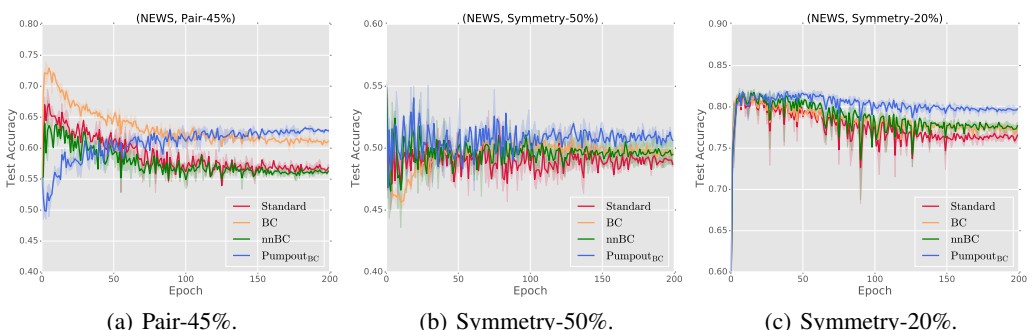

(a) Pair-45%.          (b) Symmetry-50%.          (c) Symmetry-20%.

Figure 7: Results of Pumpout_BC and nnBC on *NEWS* dataset. Test accuracy vs number of epochs.

## 5 CONCLUSION

This paper presents a meta algorithm called Pumpout, which significantly improves the robustness of state-of-the-art methods under noisy labels. Our key idea is to squeeze out the negative effects of noisy labels actively from the training model, instead of passively forgetting these effects. The realization of Pumpout is to train deep neural networks by stochastic gradient descent on "fitting" labels; while train deep neural networks by scaled stochastic gradient ascent on "not-fitting" labels. To demonstrate the efficacy of Pumpout, based on MentorNet and Backward Correction, we design two upgraded versions called Pumpout_SL and Pumpout_BC, respectively. The experimental results show that, both updated approaches can train deep models more robustly over previous ones. In future, we can extend our work in the following aspects. First, we can leverage Pumpout approach to train deep models under another weak supervision, e.g., complementary labels (Ishida et al., 2017). Second, we should investigate the theoretical guarantees for Pumpout approach. Third, we should adapt our Pumpout approach to several much harder noise cases, for example, instance-dependent noise datasets (i.e., *Open Images* (Veit et al., 2017) or *Clothing1M* (Xiao et al., 2015)).

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

# A    DEFINITION OF NOISE

The definition of transition matrix $\mathbf{T}$ is as follow, where $\epsilon$ is the noise rate and $n$ is the number of the classes.

$$\text{Pair flipping:}\quad \mathbf{T} = \begin{bmatrix} 1-\epsilon & \epsilon & 0 & \dots & 0 \\ 0 & 1-\epsilon & \epsilon & & 0 \\ \vdots & & \ddots & \ddots & \vdots \\ 0 & & & 1-\epsilon & \epsilon \\ \epsilon & 0 & \dots & 0 & 1-\epsilon \end{bmatrix},$$

$$\text{Symmetry flipping:}\quad \mathbf{T} = \begin{bmatrix} 1-\epsilon & \frac{\epsilon}{n-1} & \dots & \frac{\epsilon}{n-1} & \frac{\epsilon}{n-1} \\ \frac{\epsilon}{n-1} & 1-\epsilon & \frac{\epsilon}{n-1} & \dots & \frac{\epsilon}{n-1} \\ \vdots & & \ddots & & \vdots \\ \frac{\epsilon}{n-1} & \dots & \frac{\epsilon}{n-1} & 1-\epsilon & \frac{\epsilon}{n-1} \\ \frac{\epsilon}{n-1} & \frac{\epsilon}{n-1} & \dots & \frac{\epsilon}{n-1} & 1-\epsilon \end{bmatrix}.$$

# B    NETWORK STRUCTURES

For *MNIST*, 28×28 gray image, the structure is as follows. We also summarize it into Table 2.

(1*28*28)-[C(3*3,128)]*2-maxpool(2*2,2)-dropout(0.25)-[C(3*3,256)]*3-maxpool(2*2,2)-dropout(0.25)-C(3*3,512)-C(3*3,256)-C(3*3,128)-avgpool(1*1)-128-10, where the input is a 28*28 image, C(3*3,128) means 128 channels of 3*3 convolutions followed by LReLU (negative slop=0.01), maxpool(2*2,2) means max pooling (kernel size=2, stride=2), avgpool(2*2) means average pooling (kernel size=2), [.]*n means n such layers, etc. Batch normalization was applied before LReLU activations.

Table 2: 9-layer CNN used in our experiments on *MNIST*.

| CNN on *MNIST* |
| --- |
| 28×28 Gray Image |
| 3×3 conv, 128 LReLU |
| 3×3 conv, 128 LReLU |
| 3×3 conv, 128 LReLU |
| 2×2 max-pool, stride 2 |
| dropout, $p = 0.25$ |
| 3×3 conv, 256 LReLU |
| 3×3 conv, 256 LReLU |
| 3×3 conv, 256 LReLU |
| 2×2 max-pool, stride 2 |
| dropout, $p = 0.25$ |
| 3×3 conv, 512 LReLU |
| 3×3 conv, 256 LReLU |
| 3×3 conv, 128 LReLU |
| avg-pool |
| dense 128→10 |

For *CIFAR-10*, 32×32 RGB image, the structure is ResNet-32.

For *NEWS*, the structure is 3-layer. Batch-normalization was applied before Softsign activations. We also summarize it into Table 3.

Table 3: 3-layer MLP used in our experiments on *NEWS*.

| MLP on *NEWS* |
| --- |
| 300-D Embedding |
| dense 300→300, Softsign |
| dense 300→300, Softsign |
| dense 300→2 |

