# OpenReview forum: "Pumpout: A Meta Approach for Robustly Training Deep Neural Networks with Noisy Labels"
_ICLR.cc/2019/Conference_

### Official Review · AnonReviewer2 · 2018-10-25
**Non convincing experiments**

**Rating:** 3
**Confidence:** 4

**Review:**

The paper proposes a meta algorithm to train a network with noisy labels.
It is not a general algorithm but a simple modification of two proposed methods.  It is presented as a heuristics and it  would be helpful to derive a theoretical framework or motivation for the proposed algorithm.

My main concern is related to the experiment results. The results of the baseline method look strange. Why there is a strong decrease in the MNIST test accuracy after 20 epochs? Standard training of neural network is very robust to label noise.  In case of  20% symmetric error  (figure 2c)  the performance degradation using standard training should be very small.
Hence it is difficult to evaluate to performance of the proposed method.
At the beginning of the experiment section you mentioned several algorithms   for training with noisy labels.  I expect to compare your results to at least one of them.

---

> ### Author Response · Authors · 2018-11-22
> **My main concern is related to the experiment results. The results of the baseline method look strange. Why there is a strong decrease in the MNIST test accuracy after 20 epochs?**
>
> We are sorry your concerns. However, the degradation of the test accuracy in the early stage of training is due to the memorization effects of deep networks. Namely, the model first learns the simple and general patterns of the real data before over-fitting and memorizing the noise (which results in decreasing test accuracy). This observation has been well demonstrated by a lot of researchers recently (e.g. [1-3]), and this observation has become a common sense. From Figure 7(b) of [2] and Figures 3, 5, 6 of [3], we can clearly see the same phenomenon of memorization effects of deep networks on noisy labels.
>
> Actually, the degradation of Figure 2(c) is small. Please note that the lowest Y-axis is 0.88. The performance degradation of this case is about 0.1, which is similar to [2] and [3].
>
> By the way, to the best of our knowledge, we are not aware of a top conference/journal paper in this area claiming that "Standard training of neural network is very robust to label noise." Could you please point out a paper for our reference?
>
> References:
> [1] Zhang C, et al. Understanding deep learning requires rethinking generalization. In ICLR, 2017.
> [2] Arpit D, et al. A closer look at memorization in deep networks. In ICML, 2017.
> [3] Han B, et al. Co-teaching: Robust training of deep neural networks with extremely noisy labels. In NeurIPS, 2018.

---

> > ### Author Response · Authors · 2018-11-22
> > **At the beginning of the experiment section you mentioned several algorithms for training with noisy labels. I expect to compare your results to at least one of them.**
> >
> > Thanks for the comments. In this paper, we introduce a meta algorithm to actively squeeze out the negative effects of noisy labels from training model, instead of passively forgetting these effects. We consider how our meta algorithm can simultaneously benefit different orthogonal algorithms. Thus, in experimental section, we evaluate the efficacy of our meta algorithm Pumpout by applying it to the representative method of training on selected instances (MentorNet [1]), and the representative method of estimating the noise transition matrix (Backward Correction [2]). Note that, the choice of two baselines is to justify whether Pumpout can benefit representative state-of-the-art algorithms. The readers are encouraged to upgrade other methods by using Pumpout.
> >
> > References:
> > [1] Jiang L, Zhou Z, Leung T, et al. MentorNet: Learning data-driven curriculum for very deep neural networks on corrupted labels. In ICML, 2018.
> > [2] Patrini G, Rozza A, Menon A, et al. Making deep neural networks robust to label noise: A loss correction approach. In CVPR, 2017.

---

### Official Review · AnonReviewer1 · 2018-10-26
**Original idea with promising experimental results, but a limited contribution**

**Rating:** 5
**Confidence:** 5

**Review:**

A new method for defending against label noise during training of deep neural networks is presented. The main idea is to “forget” about wrongly labeled examples during training by an ascending step in the gradient direction. This is a meta-approach that can be combined with other methods for noise robustness; the detection of the noisy examples is specific of the base method utilised. Experimental results are promising.

In general, I find the idea original, and of potential practical use, but I believe the contribution of paper to be limited and not well supported by experiments. Moreover, I think that some claims made in this work are poorly justified.

== Method

The paper would greatly benefit from some theoretical backing of the proposed optimization scheme, even on a simplified scenario. An idea would be to prove that, given a dataset with noisy labels, PumpOut converges close to the best model (= the one learned without noise), for certain hyperparameters. I think this would be new and interesting. A result of similar fashion was proven in [A].

I found the following arguments not well or only heuristically supported:
* section 2: the scaling factor \gamma. Why using \gamma=1 is suboptimal? One could claim that as much as you want to memorize the true image-label patterns, you also want to forget the image-noise ones.
* Why MentorNet + PumpOut does not suffer from the selection bias effect of CoTraining? This is unclear to me
* The non-negative version of the BackwardCorrection, and its appeal to Kiryo et al 17 is interesting, but it sidesteps its justification. A loss that can be negative does not necessarily means that it not lower-bounded. In fact, for a minimization problem to be well-defined, all you need is a lower bounded objective. Then, adding the lower bound makes your loss non-negative. Notice that Patrini et al 17 did not state that BC is unbounded, but only that it can be negative. Can you show more that that -- maybe, at least experimentally?

The statement of Theorem 2 is trivial. In fact, no proof is given as it would be self-evident. Moreover, the Theorem is not used by PumpOut. Algorithm 3 uses a if-else conditional on the scale of the backward correction, without the max(0, .). I suggest to remove this part. I have notice later that the non-negative version of BC is used as a baseline in the experiment, but I think that is the only use.

Regarding the presentation, in section 3, I suggest to move the explanation of MentorNet and BackwardCorrection before their upgrade by PumpOut.

== Experiments

Table 1 can be removed as these are extremely common datasets.

The experimental results look very promising for applications. As a side effect of this analysis, I can also notice an improvement over BC given to the nnBC, which is nice per se. Although, I would have strengthen the empirics as follow.
* SET2 is only run on MNIST. Why not even on CIFAR10 which is used in SET1? Any future reader will wonder “did it work on CIFAR10?"
* A much harder instance of noise, for instance open set [B] or from a real dataset [Xiao et al 15] would have more clearly supported the use of PumpOut for real applications.
* Can the authors elaborate on “the choices of \beta and \gamma follows Kirkyo et al 2017” ? And how assuming their knowledge gives a fair comparison to BC which does not require them? I believe this is a critical point for the validity of the experiments.

Minor:
* “LRELU active function’ -> activation function. What is a LRELU? LeakyReLU?

[A] Malach, Eran, and Shai Shalev-Shwartz. "Decoupling" when to update" from" how to update"." Advances in Neural Information Processing Systems. 2017.
[B] Veit, Andreas, et al. "Learning From Noisy Large-Scale Datasets With Minimal Supervision." CVPR. 2017.

---

> ### Author Response · Authors · 2018-11-22
> **Section 2: the scaling factor $\gamma$. Why using $\gamma$=1 is suboptimal?**
>
> We conduct the experiments on several benchmark datasets, including vision and text. We empirically find that the best performance is usually chosen when $\gamma$ is in-between 0 and 1 by using the validation set.

---

> > ### Author Response · Authors · 2018-11-22
> > **Why MentorNet + Pumpout does not suffer from the selection bias effect of CoTraining? This is unclear to me.**
> >
> > We are sorry for such confusing. In the high level, sample selection bias means that we assign different weights to different samples, and such bias will prioritize some of samples during training. Thus, both MentorNet and $\rm Pumpout_{SL}$ (MentorNet + Pumpout) have sample selection bias. We leverage sample selection bias to overcome the label noise issue.

---

> > > ### Author Response · Authors · 2018-11-22
> > > **The non-negative version of the Backward Correction, and its appeal to Kiryo et al 17 is interesting, but it sidesteps its justification. A loss that can be negative does not necessarily means that it not lower-bounded.**
> > >
> > > We agree that a well-defined minimization problem usually has a lower bounded objective. However, a sufficient optimization on such a bounded objective does not mean a better generalization on the test set, due to the gap between the true Risk Minimization (RM) and Empirical Risk Minimization (ERM). Previous work [1] provides a counterexample that a negative but lower bounded objective function can still result in terrible over-fitting issue in PU learning. We follow a similar motivation and introduce nnBC, and present an aggressive version of nnBC ($\rm Pumpout_{BC}$) by using Pumpout.
> > >
> > > For the standard classification problem with cross-entropy loss, the training objective should be lower bounded by zero. However, when we conduct Backward Correction on cross-entropy loss for noisy labels, the corrected cross-entropy loss may become negative, which will arise the over-fitting issue. Therefore, we should bound the objective of corrected cross-entropy loss with a $\max$ operator (Theorem 2), which can avoid over-fitting issue. To achieve better performance, we extend Theorem 2 to Algorithm 3 ($\rm Pumpout_{BC}$) that actively forget negative effects of noisy labels.
> > >
> > > References:
> > > [1] Kiryo R, Niu G, du Plessis M C, et al. Positive-unlabeled learning with non-negative risk estimator. In NeurIPS, 2017.

---

> > > > ### Author Response · Authors · 2018-11-22
> > > > **The statement of Theorem 2 is trivial. In fact, no proof is given as it would be self-evident. Moreover, the Theorem is not used by PumpOut. Algorithm 3 uses a if-else conditional on the scale of the backward correction, without the max(0, .).**
> > > >
> > > > Thanks for your comments. We have modified the statement of Theorem 2. Non-negative version of BC is closely related to BC and $\rm Pumpout_{BC}$ (Algorithm 3). The corrected loss by Backward Correction can be negative, which yields over-fitting issue. Non-negative version of BC is a strategy to overcome the over-fitting issue caused by the negative loss. However, the non-negative version of BC is passive, since max operator means stopping gradients computation on negative-risk instances. $\rm Pumpout_{BC}$ is an aggressive version of non-negative BC. $\rm Pumpout_{BC}$ conducts not only stochastic gradient descent on non-negative-risk instances (when $ \mathbf{1}^{\top}\mathbf{T}^{-1}\ell(\mathbf{x}, y; w_f)\geq0$), but also scaled stochastic gradient ascent on negative-risk instances (when $\mathbf{1}^{\top}\mathbf{T}^{-1}\ell(\mathbf{x}, y; w_f)\leq0$). As non-negative BC is highly related to BC and our $\rm Pumpout_{BC}$, we present it on the text and use it as a baseline. Note that, in Algorithm 3, we set a tuning "safety valve" $\beta$, and use the fitting condition as $ \mathbf{1}^{\top}\mathbf{T}^{-1}\ell(\mathbf{x}, y; w_f)\geq \beta$. However, in all experiments, we directly set $\beta = 0$.

---

> > > > > ### Author Response · Authors · 2018-11-22
> > > > > **Regarding the presentation, in section 3, I suggest to move the explanation of MentorNet and BackwardCorrection before their upgrade by PumpOut.**
> > > > >
> > > > > Thanks for your comments. The background of MentorNet and Backward Correction has been moved before their upgraded version via Pumpout. Please check the new organization of Section~3. We have modified the Table 1 by introducing the details of two vision datasets (\textit{MNIST} and \textit{CIFAR-10}) and one text dataset (\textit{NEWS}), and added the description of the activation function "Leaky ReLu (LReLU) activation function".

---

> > > > > > ### Author Response · Authors · 2018-11-22
> > > > > > **SET2 is only run on MNIST. Why not even on CIFAR10 which is used in SET1? Any future reader will wonder "did it work on CIFAR10"?**
> > > > > >
> > > > > > Thanks for your comments. For SET2, we have added experimental results on another vision dataset \textit{CIFAR-10} and a text dataset \textit{NEWS}. Due to the limited time, we only focus on these standard benchmark datasets at current stage, and we will remain the evaluation on \textit{Open Images} dataset or \textit{Clothing1M} dataset as a future work. For $\beta$, we directly set it to zero and $\gamma$ is chosen among $\{0, 0.001, 0.005, 0.01, 0.05, 0.1, 0.5, 1\}$ via a validation set.

---

### Official Review · AnonReviewer3 · 2018-11-04
**This paper presents a meta algorithm to improve the robustness of learning methods under noisy labels.**

**Rating:** 6
**Confidence:** 3

**Review:**

This paper presents a meta algorithm to improve the robustness of learning methods under noisy labels. The idea is to squeeze out the negative effects of noisy labels actively. The paper trains deep neural networks by stochastic gradient descent on “fitting” labels; while trains deep neural networks by scaled stochastic gradient ascent on “not-fitting” labels. Experimental results show the improvement on robustness.

The good things of the paper are clear.
1.	Technical sound with reasonable idea
2.	Problem is well motivated
3.	Paper is general well written.

Some comments
1.	The idea using instance selection is not new. The novelty could be improved. If the paper could make more insight from either theoretical or application value, would be more interesting.
2.	Experiments are too standard. More divers and various data sets would be more convincing.

---

> ### Author Response · Authors · 2018-11-22
> **The idea using instance selection is not new. The novelty could be improved.**
>
> Thanks for your suggestion. Recently, in the area of deep learning with noisy labels, training on selected instances is a totally new direction, which attracts a lot of attention [1-3]. However, once some false-positive instances (real noisy data) are selected during training, DNNs will memorize them finally, which inevitably degrades the generalization performance (i.e., test accuracy) in the test phase. The key novelty of this paper is how to actively mitigate memorizing the negative effects of noisy labels, instead of following the path of training on selected instances.
>
> Specifically, to address such an issue, we introduce a meta approach called Pumpout. Intuitively, it squeezes the negative effects of noise labels actively by scaled stochastic gradient ascent. We can leverage Pumpout to upgrade orthogonal methods, such as MentorNet [1] (training on selected instances) and Backward Correction [4] (estimating the noise transition matrix).
>
> To the best of our knowledge, it is the first attempt in deep learning that studies "how to forget memorized information in an active manner", which is critical to improve the performance of some existing state-of-the-art methods. Nevertheless, we agree that some theoretical justification for Pumpout can be useful to understand and improve our method. We leave this in future works.
>
> References:
> [1] Jiang L, Zhou Z, Leung T, et al. MentorNet: Learning data-driven curriculum for very deep neural networks on corrupted labels. In ICML, 2018.
> [2] Ren M, Zeng W, Yang B, Urtasun R. Learning to reweight examples for robust deep learning. In ICML, 2018.
> [3] Han B, Yao Q, Yu X, Niu G, et al. Co-teaching: Robust training of deep Neural networks with extremely noisy labels. In NeurIPS, 2018.
> [4] Patrini G, Rozza A, Menon A, et al. Making deep neural networks robust to label noise: A loss correction approach. In CVPR, 2017.

---

> > ### Author Response · Authors · 2018-11-22
> > **Experiments are too standard. More divers and various data sets would be more convincing.**
> >
> > Thanks for your suggestion. We have added one text dataset called NEWS in updated experiments. As can be seen in Figure 4 and Figure 7, obvious improvements (similar to other figures) are achieved compared to baseline approaches, which confirms the benefits of Pumpout. Namely, Pumpout indeed can overcome the issue of memorizing noisy labels.

---

### Public Comment · ~Xiyu_Yu1 · 2018-09-28
**Simple and smart idea**

Pumpout gives an smart option for how to use information of noisy labels. Traditionally, we try to ignore the information by using data cleansing or robust reweighting methods. In this paper, the authors proposed using a scaled stochastic gradient ascent direction (instead of the normal gradient descent direction) as a more possibly correct update direction for noisy data. This idea is simple but reasonable and smart.

---

> ### Author Response · Authors · 2018-10-01
> **Your understanding is correct**
>
> Yes, your understanding is correct. The idea of our Pumpout is to use scaled stochastic gradient ascent to actively squeezes out the negative effects of noisy labels from the training model. The realization is simple and general, which will benefit orthogonal techniques in the area of deep learning with noisy labels.

---

### Public Comment · (anonymous) · 2018-10-01
**Interesting idea**

Interesting idea. It is expected to benefit the area of learning from noisy labels and it is also meaningful for real scenario applications where clean labels are not available. Hope to see the released code soon.

---

### Public Comment · (anonymous) · 2018-10-01
**Related Work**

This paper has many similarities to [1]. Superficially, Figure 1 is identical to [1]'s figure. Algorithmically, the method is highly similar too. The authors acknowledge the algorithmic similarity, but they compare to MentorNet and do little to show an improvement over [1].
Also, the method requires a label noise estimate. They say "We assume the noise level \epsilon is known and... If \epsilon is not known in advance, \epsilon can be inferred using validation sets." In practice, \epsilon needs to be estimated by manually labeling some examples. But if a small set of clean data is available, then the authors should compare to the label corruption techniques of [2] or [3] since these approaches assume access to a small trusted set of examples.
[1] Bo Han, Quanming Yao, Xingrui Yu, Gang Niu, Miao Xu, Weihua Hu, Ivor Tsang, Masashi Sugiyama. Co-teaching: Robust Training Deep Neural Networks with Extremely Noisy Labels. NIPS, 2018.
[2] Dan Hendrycks, Mantas Mazeika, Duncan Wilson, Kevin Gimpel. Using Trusted Data to Train Deep Networks on Labels Corrupted by Severe Noise. NIPS, 2018.
[3] Mengye Ren, Wenyuan Zeng, Bin Yang, Raquel Urtasun. Learning to Reweight Examples for Robust Deep Learning. ICML, 2018.

---

> ### Public Comment · ~Hassam_Sheikh1 · 2018-10-01
> **Question about unpublished but accepted papers**
>
> I see that you have mentioned 2 NIPS paper that are accepted at this years' NIPS which has not held which means that these accepted papers are not officially published yet and are only available at ArXiV. is it ok to compare a work which is not officially published as of now?

---

> > ### Public Comment · (anonymous) · 2018-10-01
> > **Reply**
> >
> > Their paper builds on a NIPS 2018 paper. If your question is about the ICLR review process, then they probably consider ICML 2018 "prior art" but not necessarily NIPS 2018 papers except for NIPS 2018 papers authors choose to cite. Hence "[2] or [3]" not "[2] and [3]."

---

> > > ### Author Response · Authors · 2018-10-05
> > > **Both [2] and [3] are great papers**
> > >
> > > Both [2] and [3] are great papers, we will definitely cite them in the suitable place later. When our experiments involve the \epsilon estimation, we will compare them and conduct the analysis.

---

> ### Author Response · Authors · 2018-10-05
> **Pumpout and Co-teaching are totally different**
>
> The differences between Pumpout and Co-teaching [1] is obvious and significant. The idea of Co-teaching [1] is to train two deep neural networks, and each network samples small-loss instances to update the parameters of its peer network.
>
> However, Pumpout is a meta approach, which aims to benefit orthogonal algorithms in deep learning with noisy labels (i.e., MentorNet, Co-teaching, Backward Correction etc.). The idea of Pumpout is to actively squeeze out the negative effects of noisy labels from the training model, instead of passively forgetting these effects. Specifically, Pumpout conducts stochastic gradient descent on “fitting” labels; and conducts scaled stochastic gradient ascent on ‘not-fitting’ labels instead of stopping gradient computation. The “fitting” labels are not limted to be the selected clean labels.
>
> To verify the efficacy of Pumpout, we leverage Pumpout to upgrade two representative but orthogonal approaches: MentorNet and Backward Correction. Note that, Co-teaching shares the similar direction with MentorNet. Thus, we do not need to compare it with [1] here. The potential comparison in future should between “Co-teaching” and “Co-teaching + Pumpout”.
>
> Moreover, we will add citation to Figure 1 in the updated version. For [2] and [3], we will cite them in our updated version, when our experiments involve the \epsilon estimation.

---

> ### Author Response · Authors · 2018-10-05
> **The experimental comparisons are fair**
>
> Sorry, we should make our point clearer about how to estimate the noise level in practice. It's true the noise level can be estimated from clean validation data. However, note that clean data require domain experts to label, and thus a small set of clean data can be as costly as a huge set of noisy data. On the other hand, we could give a representative sub-sampling of our noisy data and directly ask the domain expert to estimate the noise level. In this way, we won't obtain a small set of clean data from the domain expert. As a result, we could pay much less to only obtain some key parameters in the underlying data generation/corruption process. This is the scenario of the current paper. Therefore, the experimental comparisons are fair.

---

### Public Comment · (anonymous) · 2018-10-01
**Question about your experiments with CIFAR-10**

I read your paper on Pumpout, and I think it offers an interesting new perspective on training DNNs with noisy labels. However, I have a question about your experiments on CIFAR-10. From Figure 3 (c) of your paper.  It seems that the performance of your "Normal" baselines using Resnet-32 is not as good as baselines by recent papers ([1][2][3][4]) which did experiments with similar architectures (resnet-32 and resnet-44). Do you have any explanation for why this might have happened?

[1] Patrini, Giorgio, et al. "Making deep neural networks robust to label noise: A loss correction approach." Proc. IEEE Conf. Comput. Vis. Pattern Recognit.(CVPR). 2017.
[2] Zhang, Zhilu, and Mert R. Sabuncu. "Generalized Cross Entropy Loss for Training Deep Neural Networks with Noisy Labels." arXiv preprint arXiv:1805.07836 (2018).
[3] Ma, Xingjun, et al. "Dimensionality-Driven Learning with Noisy Labels." arXiv preprint arXiv:1806.02612 (2018).
[4] Tanaka, Daiki, et al. "Joint optimization framework for learning with noisy labels." arXiv preprint arXiv:1803.11364 (2018).

---

> ### Author Response · Authors · 2018-10-05
> **The reason comes from Data Augmentation**
>
> The main reason is that training strategies are very different. All above works [1-4] use SGD with a momentum of 0.9, weight decay of 0.0001, and data augmentation. Note that, data augmentation has been widely used in computer vision community, which significantly improves the classification performance of a deep learning model.
>
> However,  in our machine learning paper, our focus is to explore the efficacy of Pumpout. Therefore, we use Adam optimizer in all experiments for fair comparison without using data augmentation trick.
>
> For Figure 3 (c), we have also tested the performance of "Normal" baseline using ResNet32 model, and the training strategy follows above works [1-4] using data augmentation. We achieved a similar test accuracy of 81%.

---

> > ### Public Comment · (anonymous) · 2018-10-06
> > **missing comparison with dropout**
> >
> > In [5], their experiments at Table 2 (https://openreview.net/pdf?id=r1Ddp1-Rb ) using dropout (droprate=0.7) achieves > 89% test accuracy with the same setting as Figure 3.c (20% symmetric noise) though different architecture. Still, I think dropout should be compared as a strong baseline, especially since you also used it in training your models.
> >
> > [5] Zhang, Hongyi, et al. "mixup: Beyond Empirical Risk Minimization" International Conference on Learning Representations (ICLR). (2018).

---

### Author Response · Authors · 2018-11-26
**Summary of Changes**

Dear Area Chair and Anonymous Reviewers,

On behalf of all co-authors, we appreciate your great efforts in our paper review. Except our point to point response to each reviewer (see details in following posts), we hope to highlight several important points that we revised in the high level.

1. To further justify our Pumpout idea empirically,  we add a text dataset called NEWS, and conduct the corresponding experiments on Pumpout_{SL} and Pumpout_{BC}. This dataset is very important to justify that our Pumpout can be leveraged not only in vision tasks but also in text tasks, and various datasets will be more convincing (AnonReviewer1 and AnonReviewer3).

2. To remedy the concerns to our experiments, we add a lot of detailed explanations in Section 4. For example, 1) "Note that, the focus of our paper is to explore the efficacy of Pumpout. Therefore, we use Adam optimizer in all experiments for fair comparison without using data augmentation trick (Zhang & Sabuncu, 2018; Ma et al., 2018)."; 2) "Note that, the choice of two baselines is to justify whether Pumpout can benefit representative state-of-the-art algorithms. The readers are encouraged to upgrade other methods, such as Reed et al. (2015), Goldberger & Ben-Reuven (2017), and Kiryo et al. (2017) by using Pumpout.". We just showcase 2 points here, and more explanations have been merged into the revised paper.

3. To make readers easily know our algorithms, we changed the structure of Section 3. We first introduce the background of MentorNet (Backward Correction). Then, we propose  our Pumpout_{SL} (Pumpout_{BC}). Lastly, we explain the relations between MentorNet (Backward Correction) and Pumpout_{SL} (Pumpout_{BC}).

To sum up, we try our best to revise the whole paper, and hope reviewer can feed us more suggestive comments in order to make our paper better. Many thanks for all your efforts!!!

Regards,
The authors

---

### Meta-Review · Area_Chair1 · 2018-12-14

**Confidence:** 4
**Recommendation:** Reject

**Metareview:**

The paper presents an approach to mitigate the presence of noisy labels during
training by trying to forget wrong labels. Reviewers pointed out a few
concerns, including lack of novelty, lack of enough experimental support, and
lack of theoretical support. Authors have added some experiments and details
about the experimental section, but reviewers still think it's not enough
for acceptance. I concur with the reviewers to reject the paper.